

# Hawkeye: Discovering and Grounding Implicit Anomalous Sentiment in Recon-videos via Scene-enhanced Video Large Language Model

Jianing Zhao
jnzhao1106@stu.suda.edu.cn
School of Computer Science and
Technology, Soochow University
Suzhou, China

Jingjing Wang*
djingwang@suda.edu.cn
School of Computer Science and
Technology, Soochow University
Suzhou, China

Yujie Jin
yjjin0727@stu.suda.edu.cn
School of Computer Science and
Technology, Soochow University
Suzhou, China

Jiamin Luo
20204027003@stu.suda.edu.cn
School of Computer Science and
Technology, Soochow University
Suzhou, China

Guodong Zhou
gdzhou@suda.edu.cn
School of Computer Science and
Technology, Soochow University
Suzhou, China

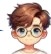

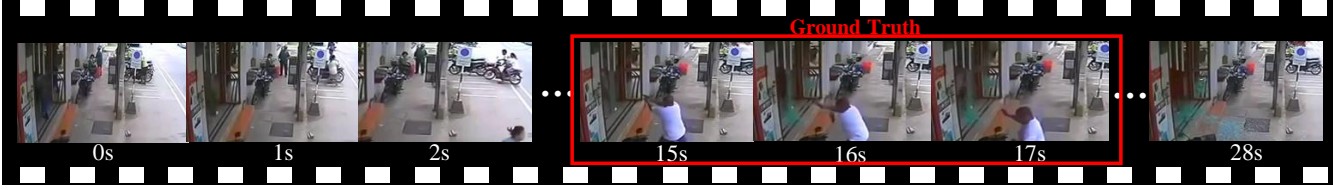

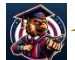

**Figure 1: An example to illustrate the IasDig task and the output of the proposed Hawkeye approach. In a recon-video like a surveillance video, Hawkeye can precisely localize frame-level fine-grained implicit anomalous sentiments (red timestamps).**

## Abstract

In real-world recon-videos such as surveillance and drone reconnaissance videos, commonly used explicit language, acoustic and facial expressions information is often missing. However, these videos are always rich in anomalous sentiments (e.g., criminal tendencies), which urgently requires the implicit scene information (e.g., actions and object relations) to fast and precisely identify these anomalous sentiments. Motivated by this, this paper proposes a new chat-paradigm **I**mplicit **a**nomalous **s**entiment **Di**scovering and **g**rounding (IasDig) task, aiming to interactively, fast discovering and grounding anomalous sentiments in recon-videos via leveraging the implicit scene information (i.e., actions and object relations). Furthermore, this paper believes that this IasDig task faces two key challenges, i.e., scene modeling and scene balancing. To this end, this paper proposes a new Scene-enhanced Video Large Language Model named Hawkeye, i.e., acting like a raptor (e.g., a Hawk) to discover and locate prey, for the IasDig task. Specifically, this approach designs a graph-structured scene modeling module and a balanced heterogeneous MoE module to address the above two challenges, respectively. Extensive experimental results on our constructed scene-sparsity and scene-density IasDig datasets demonstrate the great advantage of Hawkeye to IasDig over the advanced Video-LLM baselines, especially on the metric of false negative rates. This justifies the importance of the scene information for identifying implicit anomalous sentiments and the impressive practicality of Hawkeye for real-world applications.

*Corresponding Author: Jingjing Wang.

MM '24, October 28–November 1, 2024, Melbourne, VIC, Australia.
© 2024 Copyright held by the owner/author(s). Publication rights licensed to ACM.
ACM ISBN 979-8-4007-0686-8/24/10
https://doi.org/10.1145/3664647.3681407

## CCS Concepts

• **Computing methodologies → Artificial intelligence**.

## Keywords

Implicit Sentiment; Anomalous Information; Recon Videos; Scene-enhanced LLM

**ACM Reference Format:**
Jianing Zhao, Jingjing Wang, Yujie Jin, Jiamin Luo, and Guodong Zhou. 2024. Hawkeye: Discovering and Grounding Implicit Anomalous Sentiment in

Recon-videos via Scene-enhanced Large Language Model . In *Proceedings of the 32nd ACM International Conference on Multimedia (MM '24), October 28–November 1, 2024, Melbourne, VIC, Australia.* ACM, New York, NY, USA, 10 pages. https://doi.org/10.1145/3664647.3681407

## 1 Introduction

In the literature, Multimodal Affective Computing (MAC) [16, 55] routinely focuses on leveraging the explicit information (e.g., acoustic, facial expressions and language) for precisely understanding the sentiment of videos. These studies can be broadly divided into two paradigms: fusion paradigm [8, 46] and interaction paradigm [18, 41, 55]. However, in real world, there exists a vast amount of recon-videos, such as surveillance and drone reconnaissance videos, wherein the above explicit information is often missing, while these videos are always rich in negative sentiments (e.g., criminal tendencies). Under this scenario, it is urgent to leverage the implicit scene information (e.g., actions and object relations) for precisely discovering and grounding anomalous sentiments. In this paper, we define the above recon-videos involved negative sentiments, which require the implicit scene information for identifying, as the implicit anomalous sentiments.

With these in mind, we propose a new **I**mplicit **a**nomalous **s**entiment **Di**scovering and **g**rounding (IasDig) task, which leverages Video-LLMs through a new chat paradigm for implicit anomalous sentiment identification. This task aims at discovering and grounding (i.e., classifying and localizing) anomalous sentiments in recon-videos through implicit scene information, such as actions and object relations. To be specific, this IasDig task discovers and locates the anomalous sentiments segments within the recon-videos through interactions with LLM. For example, in Figure 1, a man near the door with his back to the camera is shooting at someone during the ground truth timestamps from 15s to 17s in the recon-video, where the LLM is required to locate the timestamps of implicit anomalous sentiment displays by the man. Under this circumstance, LLM should fully use the scene information for precisely identifying implicit anomalous sentiments. In this paper, we seek to leverage the advanced video comprehension capabilities of Video-LLMs (e.g., Video-LLaVA [29]) to address our IasDig task. However, current LLMs have deficiencies in scene understanding as reported by [27, 35]. In this way, we believe that leveraging Video-LLMs to comprehend implicit scene information at least faces two main challenges.

On one hand, it is challenging to model the scene information (i.e., action and object relations) in recon-videos. As shown in Figure 1, considering the recon-video with obscured facial expressions due to the camera angle, it is significantly difficult to precisely discover and localize anomalous sentiments. In this situation, taking the implicit scene information, such as action (*shooting at*) and object relations (<*man, near, door*>) into consideration can help better identifying anomalous sentiments. Therefore, how to model the scene information is rather important. Fortunately, some prior works have recognized this problem and utilize graph structures to model object relations information [10, 48]. Actually, the joints of the movements of human body can also be viewed as graph structures. Inspired by this, this paper believes that a well-behaved IasDig approach should consider leveraging graph structures to

model both the action and the object relations scene information for further enhancing the scene understanding ability of Video-LLMs.

On the other hand, it is challenging to balance the above two heterogeneous scene information during the alignment phase of Video-LLMs. Still take the recon-video in Figure 1 as an example, from 15s to 17s, the object relations information indicates a man near the door, whereas the action information depicts a man shooting at someone, which is a strong signal of the presence of anomalous sentiments. Assigning equal weight to both two types of scene information could lead to interference from object information, causing the model to mistakenly categorize this video segment as non-threatening. However, existing Video-LLMs process visual features as a whole, lacking the ability to balance the weights of individual elements. Therefore, this paper believes that a better-behaved Video-LLM approach for IasDig should further consider balancing the two heterogeneous scene information.

To this end, this paper proposes a tailored Scene-enhanced Video Large Language Model named Hawkeye, i.e., acting like a raptor (e.g., a Hawk) to discover and locate prey, for the IasDig task to tackle the above two challenges simultaneously. Specifically, this approach designs a Graph-structured Scene Modeling Module to model the scene information, wherein an **A**ction-**S**ensitive **G**raph (ASG) and an **O**bject-**R**elation sensitive **G**raph (ORG) are proposed to model the action and object relations information respectively. Furthermore, inspired by Mixture-of-Experts (MoE) [15, 19], this approach designs a **B**alanced **H**eterogeneous **MoE** Module (B-H MoE) along with a scene-balancing loss function to weight the heterogeneity between the two scene information. Furthermore, this paper constructs two scene-sparsity and scene-density IasDig datasets to evaluate the effectiveness of Hawkeye, and detailed experiments demonstrate that Hawkeye achieves significant performance improvements compared to the advanced Video-LLMs.

## 2 Related Work

### 2.1 Multimodal Affective Computing

Previous studies on Multimodal Affective Computing (MAC) leverage explicit signals to predict sentiments [20, 65]. They can be broadly divided into fusion paradigm [8, 46] and interaction paradigm [18, 41, 55]. Recent studies focus more on the latter paradigm. SELF-MM [57] leverages self-supervised learning to acquire the consistency and difference of modalities. UniMSE [18] and ConFEDE [55] introduce contrastive learning for mutimodal information representation. However, these studies mainly focus on explicit sentiment analysis. Recently, a few studies pose the Implicit Sentiment Analysis (ISA) while only focusing on language. Zhou et al. [63] propose composition mechanism to event-centric ISA. Fei et al. [12] introduce chain-of-thought to mimic the reasoning process. Different from all the above studies, we propose a new chat-paradigm Implicit anomalous sentiment Discovering and grounding (IasDig) task to identify implicit anomalous sentiments in recon-videos via implicit scene information. To our best knowledge, this is the first attempt to incorporate these scene information in MAC.

### 2.2 Video Grounding

Video Temporal Localization (VTL) [36] and Referring Video Object Segmentation (RVOS) [23] are two tasks within Video Grounding

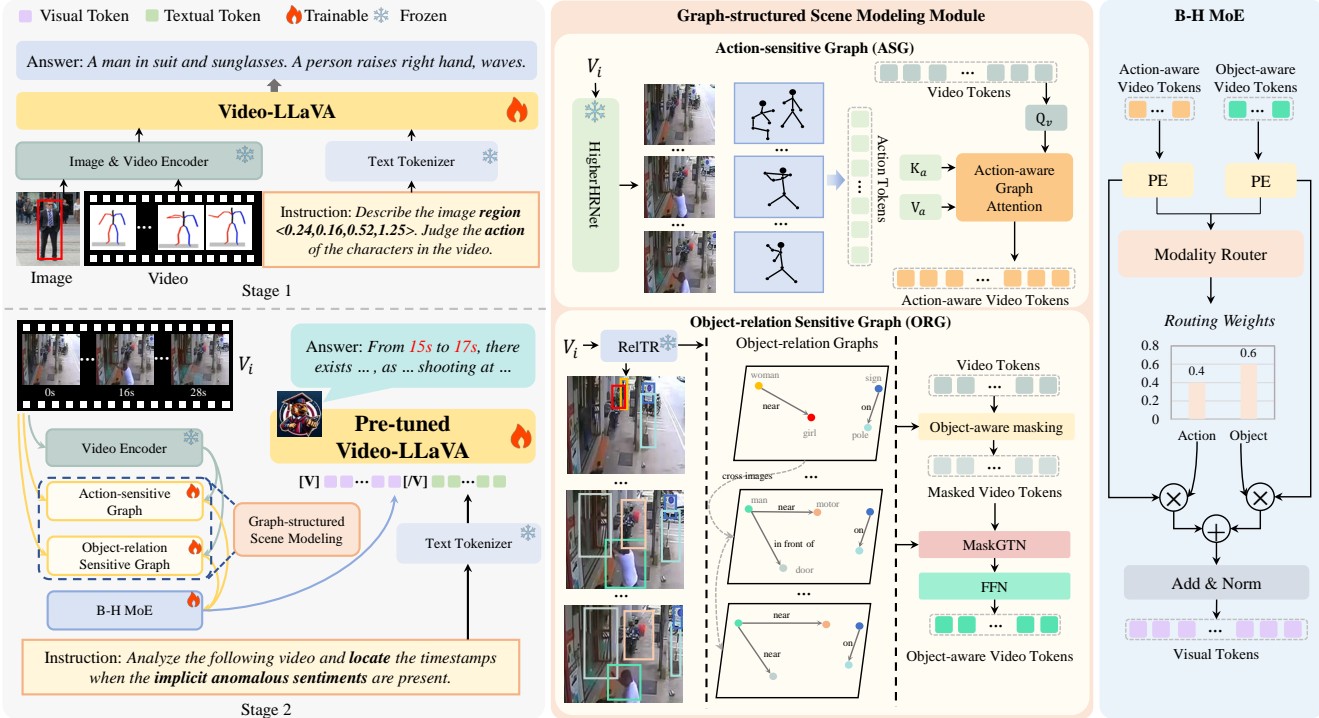

**Figure 2: The overall architecture of Hawkeye. It consists of two modules, a Graph-structured Scene Modeling Module and a Balanced Heterogeneous MoE Module (B-H MoE), where PE in B-H MoE denotes Projection Expert.**

(VG) [60]. VTL aims to utilize language query to localize temporal segments, while RVOS focuses on tracking objects correspond to the language query. For VTL, it can be broadly divided into two forms: one-stage models [42, 54] and two-stage models [30, 49, 62]. Besides, Chen et al. [4] propose an audio-enhanced approach. For RVOS, it can be divided into offline [3, 51] and online [52] processing. Particularly, Wasim et al. [50] propose an open vocabulary VG task to localize and recognize objects and actions in the video. Considering the sentiment in the video, Zhang and Yang [61] propose a temporal sentiment localization task to simultaneously localize and classify the sentiments in long untrimmed videos. Further, a relevant task for us is video-based anomaly detection [44, 53], which provides some inspiration for defining our new task, IasDig. However, existing works primarily focus on classifying the objective abnormal events, while overlook the subjective implicit anomalous sentiments especially for localizing these sentiments. Besides, these works struggle to adapt to the interactive demands of the LLM era. Different from all the above studies, this paper aims to classify and localize implicit anomalous sentiments in recon-videos, which is adapt to interactive scenarios in the era of LLMs and seek to leverage implicit scene information to perform better identification.

## 2.3 Video-oriented Large Language Models

The emergence of ChatGPT [38] has sparked a boom in the field of NLP with LLMs, such as LLaMA [45] and Vicuna [9]. Some studies [5, 6, 25, 31] expand LLMs to multimodal filed. All these studies pave the way for the development of Video-oriented Large Language Models (Video-LLMs). According to the visual encoder, Video-LLMs

can be categorized into four types. 1) VideoChat [26] and Video-LLaMA [59] utilize BLIP-2 [25] and use the Q-former to map visual representations to Vicuna. 2) Video-ChatGPT [34], Otter [24], Valley [33], mPLUG-Owl [56] and ChatUniVi [21] take advantage of CLIP [40] to obtain visual features. 3) PandaGPT [43] employs the ImageBind [13] as its backbone for video comprehension. 4) Video-LLaVA [29] leverages LanguageBind [64] to pre-align image and video features into the language feature space. However, all the Video-LLMs have overlooked the scene information in the video. Recently, a few studies [27, 35] consider incorporating scene information in images, while understanding scene in videos is almost nonexistent. Besides, some studies [15, 28] introduce the concept of MoE into LLMs, but they only focus on the efficiency, without considering the balance between different information. In summary, although all the above studies of Video-LLMs have achieved great success in video understanding, the ethical constraints limit their ability to analyze harmful content [37], therefore have difficulty tackling the IasDig task. This paper proposes a Scene-enhanced Video-LLM called Hawkeye, aiming to elicit the ability of Video-LLMs in scene understanding through scene modeling, and design a Balanced Heterogeneous MoE module to balance the scene information.

## 3 Approach

**Problem Formulation.** Given a recon-video $V$ with $T$ frames, each frame $t$ is labeled with 1 or 0, where 1 and 0 represent whether this frame conveys implicit anomalous sentiments. The goal of IasDig is to interactively discover and ground (i.e., classify and localize) the

presence of implicit anomalous sentiments in $V$, and generate a set of segments $\{(s_1, e_1), ..., (s_i, e_i), ..., (s_n, e_n)\}$, where $s_i$ and $e_i$ denote the start and end time of an implicit anomalous sentiment segment.

**Challenges.** The IasDig task faces two key challenges: scene modeling and scene balancing. To address the two challenges, we propose two modules: a Graph-structured Scene Modeling Module and a Balanced Heterogeneous MoE Module (B-H MoE).

**Backbone.** Several Video-LLMs are open-sourced currently, we consider using Video-LLaVA [29] and its visual encoder Language-Bind [64] as the backbone. It is optimized using a mixed dataset of images and videos, and achieves the best performance on most of the image and video benchmarks. Therefore, we utilize Video-LLaVA as the backbone to explore the capabilities of Video-LLMs in identifying implicit anomalous sentiments of recon-videos.

## 3.1 Graph-structured Scene Modeling Module

We design a Graph-structured Scene Modeling Module to model the implicit scene information, which consists of an **A**ction-**S**ensitive **G**raph (ASG) and an **O**bject-**R**elation sensitive **G**raph (ORG), detailed as follows.

**Action-sensitive Graph** is designed to model the action information of individuals. Specifically, we address two crucial questions: 1) how to capture the action information of individuals; 2) how to make full use of the action information in the IasDig task. We will provide comprehensive answers to these two questions in the subsequent section, formulated as follows.

For question 1), we leverage HigherHRNet[1] [7], a well-studied bottom-up human pose estimation network, to obtain the action information. As shown in ASG, given a sequence of video frames, HigherHRNet is able to generate an action graph token $x_i^a$ consisting of 17 nodes of human joints for each individual in each frame, where $i$ represents the $i$-th frame. Then, we will get the full action graph tokens $X_a = \{x_1^a, ..., x_i^a, ..., x_n^a\}$ of the video sequence[2], where $n$ represents the number of frames in this sequence.

For question 2), we leverage Action-aware Graph Attention to enrich video tokens $X_v = \{x_1^v, ..., x_i^v, ..., x_n^v\}$ from the Video Encoder with $X_a$. For each node $e_k$ in $x_i^a$, we first compute the attention weights $\alpha_{kj}$ between it and its neighboring node $e_j$:

$$\alpha_{kj} = \text{softmax}\left(\frac{(\mathbf{W}h_k) \cdot (\mathbf{W}h_j)}{\sqrt{d}}\right) \quad (1)$$

where $\mathbf{W}$ is the weight matrix, $h_k$ and $h_j$ represent the features of $e_k$ and $e_j$, and $d$ represents the dimension of features.

Then $\alpha_{kj}$ can be used to aggregate the feature $\hat{h}_k$ of node $e_k$: $\hat{h}_k = \sum_{j \in \mathcal{N}(e_k)} \alpha_{kj} \cdot h_j$, where $\mathcal{N}(e_k)$ denotes the set of neighboring nodes of node $e_k$. The final feature of $e_k$ is formulated as: $h_k' = \text{ReLU}(\mathbf{W}[\hat{h}_k, h_k])$, where $\mathbf{W}$ is the weight matrix and $[\hat{h}_k, h_k]$ denotes the concatenation of $\hat{h}_k$ and $h_k$.

After the graph attention operation, we consider enriching $X_v$ by leveraging the attention mechanism. Given the query $\mathbf{Q}_v$, key $\mathbf{K}_a$ and value $\mathbf{V}_a$, the output of a cross-attention layer is first computed as: $X_c = \mathbf{V}_a \, \text{softmax}(\mathbf{Q}_v^\top \mathbf{K}_a)$. We then borrow an idea from the transformer encoder layer [47] where each sub-layer is put into a

residual structure, and layer normalization [2] is performed after the residual connection. The input is $X_c$, and the procedure can be:

$$X_c' = \text{LN}(X_c + \mathcal{F}_{\text{MSA}}(X_c)) \quad (2)$$

$$X_{c+1} = \text{LN}(X_c' + \mathcal{F}_{\text{FFN}}(X_c')) \quad (3)$$

where $\text{LN}(\cdot)$ indicates layer normalization, $\mathcal{F}_{\text{MSA}}(\cdot)$ is the multi-head self-attention layer, and $\mathcal{F}_{\text{FFN}}(\cdot)$ represents the feed forward network. Then, we can obtain the action-aware video tokens $X_{c+1}$.

**Object-relation Sensitive Graph** is designed to model the object relations information. Specifically, there are also two questions to be answered: 1) how to effectively capture the object relations information from videos; 2) how to leverage the object relations information to help better implicit anomalous sentiment identification in the IasDig task. Next, we will answer the two questions, formulated as follows.

For question 1), we leverage RelTR[3] [10], a well-studied one-stage object-relation graph generation method, to obtain object relations information. As shown in ORG, for a given sequence of frames, RelTR is able to generate a sequence of graphs $G = \{G_1, ...G_i, ..., G_n\}$, where $G_i = (R_i, E_i)$ represents the object-relation graph of the $i$-th frame, $R_i = \{(c_{i,1}, b_{i,1}), ..., (c_{i,k}, b_{i,k})\}$ is a set of $k$ detected objects with the class $c$ and the corresponding bounding box $b$, $E_i$ denotes a set of directed edges of the form $\{c_{i,p}, r_{i,(p,q)}, c_{i,q}\}$, assigning two directional edges from $c_{i,p}$ to $r_{i,(p,q)}$ and from $r_{i,(p,q)}$ to $c_{i,q}$, where $r_{i,(p,q)}$ denotes a relationship categories. Take the last frame of the video in ORG as an example, one object can be expressed as (*man, <0.28, 0.13, 0.62, 1.36>*), and one edge can be (*man, near, door*).

For question 2), we leverage object-aware masking together with Masked Graph Transformer Networks (MaskGTN) to fully use the object relations information. We first design object-aware masking, which masks out the unimportant object parts of the frame according to the bounding box information for each frame representation $x_i^v \in X_v$. Then, we will get a sequence of masked video tokens $X_m = \{x_1^m, ..., x_i^m, ..., x_n^m\}$.

For enriching each region representation in each frame, we propose MaskGTN by following Wang et al. [48], which aggregates the information of its local neighbors through a graph transformer layer (GT) [58]. Given an input graph $G_i$ of region classes and edges, MaskGTN computes new vectors of each region and edge. Supposing that we use $L$ GTs, let $H^\ell$ be the feature representations of the $\ell$-th layer in GTs, the forward propagation becomes:

$$H^{(\ell+1)} = \sigma\left(\tilde{D}^{-\frac{1}{2}}\tilde{A}\tilde{D}^{-\frac{1}{2}}H^{(\ell)}W^{(\ell)}\right) \quad (4)$$

where $\sigma$ is an activation function on the graph, $\tilde{A}$ is the adjacency matrix of the graph $G_i$, i.e., a matrix from $E_i$, $\tilde{D}$ is the degree matrix of $\tilde{A}$, i.e., $\tilde{D}_{ii} = \sum_i \tilde{A}_{ij}$ and $W^{(\ell)}$ is a trainable weight matrix. After the last propagation, a feed forward network (FFN) will be introduced to get the final object-aware video tokens $X_s = \text{FFN}\left(H^{(L)}\right)$.

## 3.2 Balanced Heterogeneous MoE Module

After modeling the scene information, we design a Balanced Heterogeneous MoE Module (B-H MoE) for better implicit anomalous sentiment identification. In this section, we address two crucial

---

[1]https://github.com/HRNet/HigherHRNet-Human-Pose-Estimation
[2]Each token contains up to 5 people, and we average each token after graph attention.

[3]https://github.com/yrcong/RelTR

questions: 1) how to obtain the routing weights of the two scene information; 2) how to balance each type of the scene information.

For question 1), inspired by Mixture-of-Experts (MoE) [15, 19], we hope to introduce two projection experts (PE), allowing them to dynamically adjust the weights of the two heterogeneous pieces of the scene information.

As shown in Figure 2, in contrast to other methods that introduce several FFNs within LLMs, B-H MoE consists of two PEs outside the LLM, which dynamically adjust the weights of information from ASG and ORG. Each PE is a stack of transformer layers, and a dynamic *Modality Router R* is introduced to control the contribution of each PE. The router $R$ is structured as a straightforward MLP that receives input tokens and calculates the routing weights for each expert, i.e., a soft router [39]. Formally, for the output $h$ of ASG and ORG, the output of B-H MoE can be represented as follows:

$$y = \text{LN}\left(\sum_{i=1}^{N} R(h)_i E_i(h)\right) \quad (5)$$

where $R(h)_i$ and $E_i(h)$ denote the corresponding weight and the output of the $i$-th PE, respectively. $N$ is the number of PEs. The router $R(\cdot)$ can be written as: $R(\cdot) = \text{softmax}(h\mathbf{W}_g)$, where $\mathbf{W}_g$ is the trainable weight matrix for router $R(\cdot)$.

For question 2), inspired by LoRAMoE [11], we design a new balancing constraint loss $\mathcal{L}_{bc}$ to optimize the soft router in B-H MoE. By minimizing the $\mathcal{L}_{bc}$ loss, Hawkeye can encourage the router $R$ to dynamically adjust the contributions among all experts. Formally, we define the importance matrix $\mathbf{M}$, and $\mathbf{M}_{i,m}$ denotes the sum of importance value of the $i$-th PE in B-H MoE for the $m$-th training sample in a batch, which can be represented as follows:

$$\mathbf{M}_{i,m} = \sum_{j=1}^{T_m} \exp\left(\omega_i^j / \tau\right) \quad (6)$$

where $T_m$ and $\omega_i^j$ denote the token numbers of the $m$-th training sample, the hidden output of the $j$-th token. $\tau$ is a hyper-parameter.

We then define a coefficient matrix $\mathbf{P}$ with the same size of $\mathbf{M}$, corresponding to the importance matrix $\mathbf{M}$, $\mathbf{P}_{i,m}$ denotes the importance coefficient of $\mathbf{M}_{i,m}$, which can be written as follows:

$$\mathbf{P}_{i,m} = \begin{cases} R_i^a, & \text{Type}(m) = \text{Action} \\ R_i^s, & \text{Type}(m) = \text{Object} \end{cases} \quad (7)$$

where $R_i^a$ and $R_i^s$ are routing weights of the action information and the object information from HeteroMoE, Type$(m)$ is the type of the $m$-th training sample, i.e., action or object relations information.

The weighted importance matrix $\mathbf{T}$ can be $\mathbf{T} = \mathbf{P} \circ \mathbf{M}$. In order to constrain the scales of the scene information and meanwhile keep a balance between them, we introduce the Coefficient of Variation (CV) [1], and the new loss $\mathcal{L}_{bc}$ is formulated as: $\mathcal{L}_{bc} = \text{CV} = \left|\frac{\sigma^2(\mathbf{T})}{\mu(\mathbf{T})}\right|$, where $\mu(\mathbf{T}) = \frac{\sum_{i=1}^{m}\sum_{j=1}^{n} t_{ij}}{m \cdot n}$ and $\sigma^2(\mathbf{T}) = \frac{\sum_{i=1}^{m}\sum_{j=1}^{n}(t_{ij}-\mu)^2}{m \cdot n}$ represent the variance and mean of $\mathbf{T}$; $n$, $m$ are the rows and columns of $\mathbf{T}$; and $t_{ij}$ is element in the matrix $\mathbf{T}$. The variance represents the dispersion of the distribution of expert importance, while the mean provides an indication of the central tendency of the distribution. $\mathcal{L}_{bc}$ provides a dimensionless measure, allowing for comparisons between different scene information.

## 3.3 Model Optimization for Hawkeye

We observe that Video-LLaVA lacks the ability to comprehend the scene information, which is important for our study. To address this, we take a two-stage training process.

**Stage 1. Object-relation and Action Enhanced Pre-Tuning.** As shown in Figure 2, we first pre-tune Video-LLaVA using two high quality and manually annotated datasets, i.e., RefCOCO [22] and HumanML3D [14], which aims at enhancing Video-LLaVA with the scene understanding ability. The model is asked to *"Describe the image region and judge the action of the characters in the video"*.

**Stage 2. IasDig Tuning.** At the second stage, our goal is to make the model able to tackle the IasDig task through instruction tuning. We construct an IasDig-tuning dataset and the pre-tuned Video-LLaVA is asked to *"Analyze the following video and locate the timestamps when the individuals in the video convey implicit anomalous sentiments"*. The instruction will undergo Text Tokenization to obtain the textual tokens $\mathbf{X}_t$. The input of the LLM will be "$y$ from Eq.(5) + $\mathbf{X}_t$". The overall loss of Hawkeye can be represented as $\mathcal{L}_{total} = \mathcal{L} + \lambda\mathcal{L}_{bc}$, where $\mathcal{L}$ is the next-token prediction loss of LLMs, and $\lambda$ controls the strength of the balancing constraint loss.

## 4 Experimental Settings

### 4.1 Instruction Data Construction for Hawkeye

The training pipeline of Hawkeye contains two stages and each stage needs an instruction dataset, detailed as follows.

**For Stage 1.** As shown in Figure 3, to enhance the scene understanding ability w.r.t actions and object relations of Video-LLMs, we construct a dataset based on RefCOCO and HumanML3D. Specifically, based on these datasets, we manually construct the instruction for each video, for instance: *Instruction: "Describe the image region <objs> and judge the action of the characters in the video." Answer: "A man in suit wearing sunglasses. & A person raises right hand, waves.",* where <objs> denotes the coordinates of the region. As HumanML3D has 25K videos with an average duration of 1 seconds, and we take 8 frames per second. For the data balance, we randomly select 200K images from RefCOCO to form the dataset.

**For Stage 2.** As shown in Figure 3, we construct an IasDig-tuning dataset consisting of 1K videos. We take 8 frames per second, resulting in 1160K frames. This dataset consists of a real-word **Scene-sparsity** IasDig dataset and a real-word **Scene-density** IasDig dataset. Scene-sparsity (short for **S-S**) is based on TSL-300 [61], which includes only daily life scenes for temporal sentiment localization (thus called scene-sparsity). To fit the IasDig scenario, we have manually filtered out videos have facial expressions and removed the audio of all videos. Each video in S-S contains 0.33 anomalous segments on average. Scene-density (short for **S-D**) is based on UCF-Crime [44], a real-world surveillance video dataset containing 13 crime scenes (thus called scene-density). As IasDig mainly focuses on the anomalous sentiments, we only use the abnormal videos with negative sentiments. Each video in S-D contains 1.09 anomalous segments on average. Further, we construct the instruction for each video manually, for instance: *Instruction: "Analyze the following video and locate the timestamps when the implicit anomalous sentiments are present." Answer: "From <s1> to <s2>, there exists ..., as ... shooting at ...",* where <s1> and <s2> are the start and end time of an anomalous segment.

**Table 1: Comparison of several Video-LLMs and Hawkeye on Scene-sparsity and Scene-density Iasdig dataset. The ↓ beside FNRs indicates the lower the metric, the better the performance. I.T. means inference time in seconds for a one minute video.**

| Approaches | Scene-sparsity IasDig Dataset (S-S) | | | | | | | Scene-density IasDig Dataset (S-D) | | | | | | |
| | FNRs↓ | F2 | mAP@tIoU | | | | I.T. | FNRs↓ | F2 | mAP@tIoU | | | | I.T. |
| | | | 0.1 | 0.2 | 0.3 | Average | | | | 0.1 | 0.2 | 0.3 | Average | |
| Video Chat | 44.5 | 29.51 | 30.93 | 20.62 | 8.25 | 22.63 | 55 | 61.4 | 32.27 | 29.11 | 14.77 | 6.75 | 16.88 | 67 |
| Video ChatGPT | 61.26 | 30.66 | 25.56 | 18.89 | 10 | 18.15 | 72 | 78.48 | 17.99 | 17.02 | 6.83 | 4.25 | 9.37 | 80 |
| Valley | 56.07 | 25.77 | 31.35 | 15.15 | 6.76 | 17.75 | 63 | 68.6 | 27.51 | 28.79 | 14.14 | 7.73 | 16.87 | 62 |
| PandaGPT | 49.12 | 30.86 | 28.28 | 17.17 | 7.98 | 17.81 | 68 | 68.77 | 21.72 | 18.65 | 9.52 | 4.76 | 10.98 | 74 |
| mPLUG-Owl | 71.37 | 23.17 | 30.3 | 12.2 | 3.36 | 15.29 | 67 | 54.13 | 32.36 | 29.66 | 16.55 | 6.9 | 17.7 | 72 |
| Chat-UniVi | 61.81 | 21.78 | 18.82 | 10.61 | 9.05 | 12.83 | 89 | 80.52 | 21.65 | 16.9 | 7.75 | 2.11 | 8.92 | 92 |
| Video-LLaVA | 44.32 | 29.22 | 31.41 | 15.78 | 8.82 | 18.67 | 65 | 76.34 | 23.65 | 19.01 | 9.38 | 4.57 | 10.99 | 69 |
| Hawkeye | **35.82** | **38.09** | **35.24** | **21.21** | **14.71** | **23.72** | 66 | **45.66** | **45.03** | **34.41** | **19.22** | **12.1** | **21.91** | 74 |
| w/o ASG | 41.27 | 33.45 | 33.38 | 20.36 | 12.38 | 22.04 | 63 | 51.83 | 38.5 | 27.47 | 12.88 | 7.73 | 16.03 | 71 |
| w/o ORG | 37.47 | 35.28 | 34.69 | 21.01 | 11.52 | 22.41 | 65 | 48.88 | 39.03 | 31.82 | 12.96 | 8.33 | 17.7 | 72 |
| w/o BH-MoE | 42.94 | 31.33 | 29.59 | 18.37 | 9.18 | 19.05 | 62 | 58.79 | 34.22 | 24.33 | 12.32 | 7.38 | 14.68 | 70 |
| w/o $\mathcal{L}_{bc}$ | 41.56 | 33.95 | 33.33 | 19.78 | 12.24 | 21.78 | 60 | 47.14 | 40.43 | 25.21 | 13.45 | 8.4 | 15.69 | 68 |
| w/o Pre-tuning | 43.96 | 28.77 | 31.63 | 16.33 | 10.2 | 19.39 | 64 | 69.8 | 28.01 | 18.61 | 10.39 | 6.49 | 11.83 | 73 |

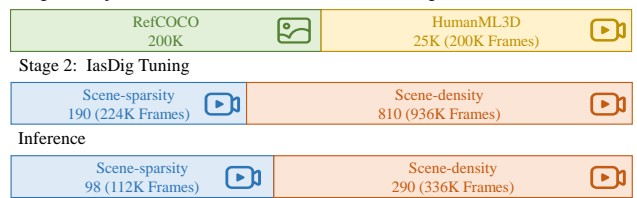

**Figure 3: Data composition for training and inference.**

**For Inference.** The test set of S-S contains 98 videos, while the test set of S-D contains 290 videos. The instructions can be the same as those in Stage 2[4].

## 4.2 Baselines

Traditional methods cannot be directly applied to the IasDig task, while general Video-LLMs can, so we choose several of them as baselines. **VideoChat** [26] utilizes Q-Former to map visual features to Vicuna [9]. **Video-ChatGPT** [34] combines LLM with CLIP [40]. **Valley** [33] uses a temporal modeling module. **PandaGPT** [43] utilizes ImageBind [13] to demonstrate cross-modal capabilities. **mPLUG-Owl** [56] uses a visual abstractor to align different modes. **Chat-UniVi** [21] merges visual tokens with semantic meanings. **Video-LLaVA** [29] conducts joint training stages on images and videos. Because the above approaches are for different tasks and have different experimental settings, for a fair and thorough comparison, we re-implement these approaches using their released codes and all the LLM size is 7B in our experiments.

## 4.3 Evaluation Metrics

IasDig focuses on discovering and grounding implicit anomalous sentiments, so it needs two evaluation metrics, detailed as follows.

For the localization performance, we adopt the commonly used mAP@tIoU metric [61]. Under different intersection over union (IoU), the metric is calculated by mean Average Precision (mAP). We set thresholds ranging from 0.1 to 0.3, with an interval of 0.1.

For the classification performance, as IasDig mainly focuses on precisely discovering the anomalous sentiments, we prefer false-negative rates (FNRs), denoting the rates of "mistaking a frame labeled positive (i.e., 1) for negative (i.e., 0)", which is of great importance for the IasDig task. For the example in Figure 1, we'd rather identify all timestamps as anomalous than miss a single timestamp with anomalous sentiment. If a frame is missed, it could result in a serious criminal event, so this is a typical FNRs problem. We also prefer Recall over Precision and report F2 [61] as another classification metric. The two classification metrics are denoted as:

$$\text{FNRs} = \frac{\# \ of \ false\text{-}negative \ frame}{\# \ of \ positive \ frame} \tag{8}$$

$$F_\beta = \frac{(1 + \beta^2) \times Precision \times Recall}{\beta^2 \times Precision + Recall} \tag{9}$$

where # denotes the frame numbers and $\beta$ is 2. Moreover, $t$-test[5] is used to evaluate the significance of the performance.

## 4.4 Implementation Details

We utilize open-source codes to obtain experimental results of all the baselines on S-S and S-D dataset. The hyper-parameters of these baselines remain the same setting reported by their public papers. The others are tuned according to the best performance. For Hawkeye, we take 8 frames per second of each video. In training stage, we use AdamW [32] as the optimizer. The initial learning rate is 2e-5, with a warmup ratio of 0.03. For stage 1 and stage 2, we fine-tune Video-LLaVA (7B)[6] using LoRA [17], with the dimension, scaling factor, dropout rate of the LoRA matrix set to be 16, 64 and 0.05, respectively, while keeping other parameters at their default values. The number of experts in B-H MoE is 2, and the layers of each expert are set to be 8. The hyper-parameters $\tau$ and $\lambda$ of $\mathcal{L}_{bc}$ are set to be 0.07 and 0.2. Hawkeye is trained for one epoch with a batch size of 8. All training runs on 1 NVIDIA A100 GPU with

---

[5]https://docs.scipy.org/doc/scipy/reference/stats.html
[6]https://github.com/PKU-YuanGroup/Video-LLaVA

**Table 2: Comparison of several well-performing Video-LLMs and Hawkeye on the 13 crime scenes of S-D dataset with FNRs.**

| Approaches | Abuse | Arrest | Arson | Assault | Accident | Burglary | Explosion | Fighting | Robbery | Shooting | Stealing | Shoplifting | Vandalism |
|---|---|---|---|---|---|---|---|---|---|---|---|---|---|
| Video Chat | 87.5 | 42.16 | 81.07 | 78.47 | 47.52 | 56.34 | 74.11 | 49.91 | 68.35 | 54.6 | 64.25 | 71.58 | 66.23 |
| Video ChatGPT | 93.75 | 70.7 | 76.62 | 94.45 | 83.67 | 78.45 | 78.9 | 71.76 | 65.42 | 70.34 | 79.95 | 87.59 | 83.12 |
| Valley | 69.5 | 49.63 | 52.86 | 79.51 | 84.16 | 55.62 | 50.35 | 50 | 100 | 92.76 | 74.88 | 88.49 | 72.73 |
| PandaGPT | 72.5 | 70.52 | 66.07 | 72.57 | 66.68 | 67.94 | 68.62 | 69.81 | 68.05 | 68.39 | 70.78 | 67.99 | 66.24 |
| mPLUG-Owl | 75 | 50.37 | 55 | 68.61 | 59.41 | 55.25 | 58.87 | 56.49 | 61.63 | 55.43 | 43 | 56.47 | 66.23 |
| Chat-UniVi | 100 | 76.87 | 87.5 | 85.42 | 94.06 | 74.64 | 87.59 | 85.06 | 72.18 | 81.06 | 45.89 | 97.48 | 80.52 |
| Video-LLaVA | 100 | 69.77 | 67.38 | 91.78 | 86.14 | 70.41 | 78.96 | 77.71 | 85.96 | 72.24 | 68.28 | 89.69 | 64.94 |
| Hawkeye | **62.5** | **37.06** | **22.68** | **47.35** | **40.92** | **34.72** | **36.29** | **43.51** | **54.14** | **39.74** | **34.3** | **52.52** | **25.11** |

40GB GPU memory. It takes around 8h for training stage 1, 40h for training stage 2 and 6h for inference.

## 5 Results and Discussions

### 5.1 Experimental Results

Table 1 presents a comparative analysis of the performance of various approaches. From this table, you can see that: **1)** The performances of Hawkeye and other Video-LLMs on both the S-S and S-D datasets are significantly poor. In cases of the best performing Video-LLMs, the average results of FNRs, F2 and mAP@tIoU are only 49.22%, 31.61%, and 20.12%. As for Hawkeye, the average results are 40.74%, 41.56%, and 22.82%. This suggests that the Ias-Dig task poses significant challenges, and the current approaches still have a lot of room for improvement in identifying implicit anomalous sentiments in real-world recon-videos. **2)** Hawkeye consistently performs better than other Video-LLMs. Compared to the best performing approach on all metrics, Hawkeye achieves average improvements of 9.95% and 2.7% on F2 and mAP@tIoU on both S-S and S-D datasets. Statistical significance tests show that these improvements are significant ($p$-value < 0.01). This demonstrates that Hawkeye can better model and balance the scene information in recon-videos compared to advanced Video-LLMs.

### 5.2 Contributions of Each Key Component

To delve deeper into the impact of the key components of Hawkeye, we conduct a series of ablation studies, the results of which are detailed in Table 1. **Effectiveness Study of Scene Information.** From Table 1, you can see that: **1) w/o ASG** exhibits inferior performance on both datasets compared to Hawkeye, with average decreases of FNRs, F2, and mAP@tIoU by 5.81% ($p$-value < 0.01), 5.59% ($p$-value < 0.01), and 3.78% ($p$-value < 0.01). This substantiates that the action information is important in implicit anomalous sentiment identification and the ASG block is efficient in modeling action information. **2) w/o ORG** also shows inferior performance on both datasets compared to Hawkeye, with average decreases of FNRs, F2, and mAP@tIoU by 2.44% ($p$-value < 0.05), 4.41% ($p$-value < 0.01), and 2.76% ($p$-value < 0.05). This justifies the importance of the object relations information and the effectiveness of ORG in modeling such information. **Effectiveness Study of BH-MoE.** From Table 1, you can see that: **1) w/o BH-MoE** exhibits inferior performance compared to **w/o ASG** and **w/o ORG** on both datasets, with average decreases of FNRs, F2, and mAP@tIoU by 4.32%, 7.69% ($p$-value < 0.01); 3.2%, 4.38% ($p$-value < 0.01); and 2.17%, 3.19% ($p$-value < 0.05), respectively. This indicates a heterogeneity between the implicit scene information. **2) w/o BH-MoE** shows inferior performance compared to **Hawkeye**, with average decreases of

FNRs, F2, and mAP@tIoU by 10.13% ($p$-value < 0.01), 8.79% ($p$-value < 0.01), and 5.95% ($p$-value < 0.01). This further demonstrates the effectiveness of BH-MoE in scene balancing and encourages us to consider handling heterogeneous issues in the manner of MoE. **3) w/o BH-MoE** shows inferior performance compared to **w/o $\mathcal{L}_{bc}$**, with average decreases of FNRs, F2 and mAP@tIoU by 6.52% ($p$-value < 0.01), 4.42% ($p$-value < 0.01) and 1.87% ($p$-value < 0.05). This further demonstrates the effectiveness of $\mathcal{L}_{bc}$ in scene balancing and encourages us to consider design a tailored loss function in the MoE module for better implicit anomalous sentiment identification. **Effectiveness Study of Pre-tuning.** From Table 1, you can see that: **w/o Pre-tuning** shows inferior performance compared to **Hawkeye**, with average decreases of FNRs, F2, and mAP@tIoU by 16.14% ($p$-value < 0.01), 13.17% ($p$-value < 0.01), and 7.21% ($p$-value < 0.01). This is reasonable and again confirms that the backbone lacks the ability to comprehend the scene information. This further demonstrates the necessity and effectiveness of Pre-tuning, and encourages us to use more high quality datasets to facilitate the scene understanding ability of Video-LLMs before tuning on IasDig.

### 5.3 Practicality Study of Hawkeye via FNRs

To study the practicality of Hawkeye, we compare the FNRs of Hawkeye with other Video-LLMs. From Table 1 you can see that: Hawkeye performs the best on the metric of FNRs. On S-S, it outperforms Video-LLaVA by 8.5% ($p$-value < 0.01), and on S-D, it outperforms mPLUG-Owl by 8.47% ($p$-value < 0.01). This indicates that Hawkeye is effective in reducing the rates of FNRs, which is of great importance in practical applications. Moreover, recognizing that the S-D dataset encompasses 13 distinct real-world crime scenes, we perform a detailed analysis of each crime scene on the performance of FNRs, which is shown in Table 2[7]. From this table you can see that: Hawkeye significantly outperforms all other Video-LLMs across the 13 anomalies. This further demonstrates the effectiveness of Hawkeye and its robust ability to reduce the FNRs, thereby reducing the likelihood of misclassifying anomalously sentimented samples as normal. This also underlines the necessity to consider the scene information in the IasDig task, especially in the case of *Vandalism*.

Besides, we also study the inference time of Hawkeye and other Video-LLMs, from Table 1, you can see that: Hawkeye does not perform much differently from the other models in terms of inference time. The inference time of Hawkeye for a one minute video is 66s on S-S and 74s on S-D. While the fastest inference time on

---

[7] As FNRs can truly reflect the practicality of the IasDig task, and limited by the length of the article, we do not illustrate F2 and mAP@tIoU on these 13 crime scenes. Actually, Hawkeye is still the best performing approach.

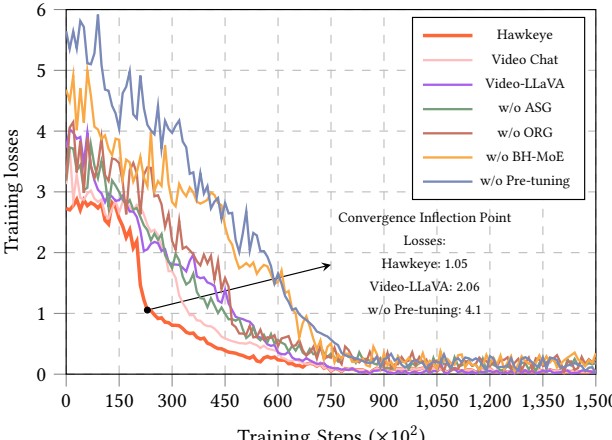

**Figure 4: Convergence of training losses across training steps.**

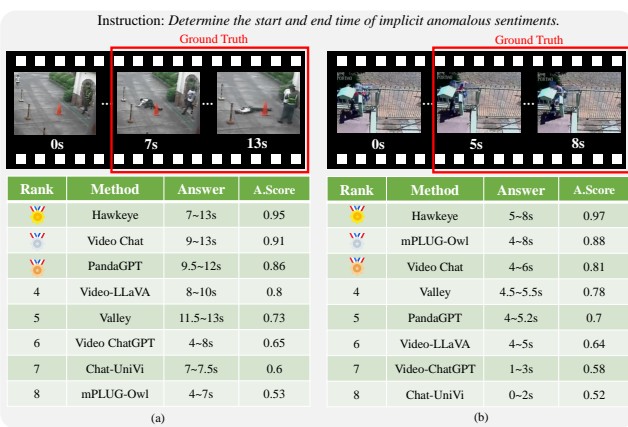

**Figure 5: Two samples to compare Hawkeye with other Video-LLMs, where A.Score means Anomaly Score.**

these two datasets are 55s (Video Chat) and 62s (Valley). This is reasonable, as some studies confirm that the MoE architecture can improve efficiency [11, 28]. This suggests that introducing more information along with a MoE module for implicit anomalous sentiment identifying does not increase the inference time and Hawkeye can maintain good inference efficiency while reducing the FNRs.

## 5.4 Convergence Analysis of Hawkeye

As illustrated in Figure 4, we study the efficacy of Hawkeye. We analyze the convergence patterns of training losses for two strong Video-LLMs (Video Chat and Video LLaVA), Hawkeye and its variant without specific components over various training steps. From this figure, you can see that: **1) Hawkeye** demonstrates the fastest convergence compared to Video Chat and Video-LLaVA. At the convergence inflection point, the loss of Hawkeye is 1.05, while Video-LLaVA is 2.06. This underscores the high efficiency of Hawkeye over other strong Video-LLMs, which hints at the potential of Hawkeye for quicker training times and less resource utilization, thereby augmenting its practical utility in real-world applications. **2) Hawkeye** demonstrates the fastest convergence compared to its variant without specific components. This justifies that the two types of scene information along with B-H MoE can accelerate the convergence process, which further encourages us to consider the scene information in the IasDig task. **3) Hawkeye** demonstrates faster convergence compared to **w/o Pre-tuning**, whose loss is 4.1 at the convergence inflection point. This justifies the importance of scene understanding before IasDig tuning and encourages us to consider using more high quality datasets for scene understanding before tuning Hawkeye on the IasDig datasets.

## 5.5 Qualitative Analysis

As shown in Figure 5, we visualize and compare the performance of Hawkeye with other Video-LLMs. We randomly select two samples from each of the S-S and S-D dataset and ask these approaches to "*Determine the start and end time of implicit anomalous sentiments*". From this figure you can see that: **1)** Predicting implicit anomalous segments is challenging. For instance, Figure 5 (b) captures a segment from the 5-*th* to the 8-*th* second where a man appears to be abusing a pet dog with a stick. Identifying his abusive behavior is

particularly challenging, necessitating the assessment of sentiments based purely on actions. **2)** Compared to other well-performing Video-LLMs, Hawkeye excels in either accuracy or predictive coverage when it comes to locating segments with implicit anomalous sentiments. In Figure 5 (a), Hawkeye excels Video Chat in predictive coverage and in (b), it excels mPLUG-Owl in accuracy. The anomaly score of Hawkeye is also the highest. This further demonstrates the effectiveness of Hawkeye in precisely locating implicit anomalous segments in recon-videos.

## 6 Conclusion

In this paper, we propose a new and challenging task, **I**mplicit **a**nomalous **s**entiment **Di**scovering and **g**rounding (IasDig), poised to significantly contribute to future research in video anomalous sentiment discovering and grounding. The advanced method Hawkeye is presented to enhance anomalous sentiment identification in recon-videos by leveraging the implicit scene information, with potential applications in maintaining social order, national defense security, etc. The core concept of Hawkeye involves utilizing two modules, the Graph-structured Scene Modeling Module and the Balanced Heterogeneous MoE Module to effectively model and balance the action and object relations information. Experimental results on our constructed scene-sparsity and scene-density IasDig datasets demonstrate the superior performance of Hawkeye over several advanced Video-LLMs. In our future work, we would like to introduce more scene information in videos (e.g., videos style and video-evolved events information) to further boost implicit anomalous sentiment identification. In addition, as shown in Table 1, the inference time is still expensive and encourages us to leverage some light-weighting technologies (e.g., LLM distillation and compression) to further improve the inference speed of our Hawkeye model.

## Acknowledgments

We thank our anonymous reviewers for their helpful comments. This work was supported by three NSFC grants, i.e., No.62006166, No.62376178 and No.62076175. This work was also supported by a Project Funded by the Priority Academic Program Development of Jiangsu Higher Education Institutions (PAPD).

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
