# OpenReview forum: "Hawkeye: Discovering and Grounding Implicit Anomalous Sentiment in Recon-videos via Scene-enhanced Video Large Language Model"
_acmmm.org/ACMMM/2024/Conference — MM2024 Oral_

### Official Review · Reviewer_Twte · 2024-05-14

**Rating:** 4
**Confidence:** 3

**Summary:**

This paper introduces a novel task—Implicit Anomalous Sentiment Discovery and Grounding (IasDig), which utilizes implicit scene information (such as actions and object relations) to identify anomalous sentiments in reconnaissance videos. The paper develops a Scene-enhanced Video Large Language Model named Hawkeye, which combines a graph-structured scene modeling module and a balanced heterogeneous MoE module to accurately discover and locate anomalous sentiments in videos. Through extensive experiments conducted on specially constructed scene-sparsity and scene-density IasDig datasets, the paper demonstrates the advantages of the Hawkeye model over advanced video LLM models in terms of accuracy and practicality.

**Strengths:**

1. The paper proposes a new IasDig task, providing a new research direction for automatic recognition of implicit anomalous sentiments in reconnaissance videos.

2. By integrating graph-structured scene modeling with a balanced heterogeneous MoE module, the paper effectively addresses the modeling and balancing of scene information.

3. The authors have constructed specific datasets and comprehensively evaluated the proposed model through comparisons with multiple advanced video LLM models.

4. The structure of the article is clear, and both the methodology and experimental design are well articulated, facilitating understanding and reproducibility.

**Limitations:**

**1. Lack of Technical Detail:** Although the paper mentions graph-structured scene modeling and balanced heterogeneous MoE modules, the descriptions of specific implementation details (such as parameter settings and algorithm optimizations) are not sufficiently detailed, which may affect the transparency and reproducibility of the model.

**2. Limitations:** Although the effectiveness of the model is validated on the custom-built datasets, it has not been tested on publicly available standard datasets, which may limit the validation of the model's generalization ability.

**3. Experimental Design:** There is a lack of comparison with traditional non-LLM methods, which could help more comprehensively showcase the advantages and innovations of Hawkeye against the existing technology backdrop.

**Suitability:**

3

---

### Official Review · Reviewer_Fj1C · 2024-05-20

**Rating:** 4
**Confidence:** 3

**Summary:**

This paper introduces Hawkeye, a novel system designed to discover and ground implicit anomalous sentiments in recon-videos.
It addresses the absence of explicit sentiment cues like language, audio by leveraging implicit scene facial expressions by leveraging implicit scene information.
The authors propose a new task, Implicit anomalous Sentiment Discovering and Grounding, and develop a Scene-enhanced Video Large Language Model (Hawkeye) to execute this task efficiently.

**Strengths:**

- This paper pioneers the use of implicit scene information for sentiment analysis in recon-videos, filling a gap where traditional sentiment analysis methods are insufficient due to the lack of explicit cues.
- The work includes extensive experiments on custom-built datasets that demonstrate Hawkeye's superiority over sota video-llms in identifying and localizing implicit anomalous sentiments.
- The Balanced Heterogeneous MoE module and scene-balancing loss function show technical sophistication in handling the complexity of scene information, enhancing model adaptability and performances.

**Limitations:**

- While the paper compares Hawkeye to several video-llms, a more comprehensive comparison with established models in related fields, such as anomaly detection or multimodal sentiment analysis, would strengthen its position.
- The paper might benefit from more detailed implementation details, especially for the B-H MoE module and scene-balancing loss, to ensure researchers can replicate and build upon the work.
- When constructing S-S and S-D datasets, what criteria were used to ensure their representativeness and diversity for evaluating Hawkeye's performance?
- Are there any plans or ongoing work to validate Hawkeye's performance on real-world recon-videos?

**Suitability:**

2

---

### Official Review · Reviewer_YAQm · 2024-05-24

**Rating:** 4
**Confidence:** 2

**Summary:**

The paper introduces a novel task called Implicit anomalous sentiment Discovering and grounding (IasDig), which is designed to identify and ground anomalous sentiments in surveillance and drone reconnaissance videos.  The paper proposes a new Scene-enhanced Video Large Language Model named Hawkeye.  The paper reports extensive experimental results on specially constructed datasets, demonstrating Hawkeye's superior performance over advanced VideoLLM baselines, particularly in reducing false negative rates.

**Strengths:**

This paper introduces a novel task and dataset.

This paper is well-written and easy to understand.

This paper has done sufficient ablation experiments to verify the proposed model.

**Limitations:**

It seems that the proposed task is similar to Video Grounding, the difference between the two tasks is only the format of the output.

Can the proposed method solve Video Grounding? Or the Video Grounding method can also solve the IasDig task? I think it is appropriate to make a comparison between the two types of methods.

Since this paper is about anomalous emotion detection, why do we need to capture actions? From this perspective, this task is very similar to Temporal Grounding of Human Motion. I don’t quite understand the difference between tasks.

**Suitability:**

3

---

### Official Review · Reviewer_PJXi · 2024-05-26

**Rating:** 4
**Confidence:** 3

**Summary:**

- The paper proposes a new Scene-enhanced Video Large Language Model named Hawkeye for the task of discovering and grounding anomalous sentiments in recon-videos (IasDig). The proposed method consists of two modules: a graph-structured scene modeling module and a balanced heterogeneous MoE module, designed to address the tasks of scene modeling and scene balancing in the IasDig task. Extensive experiments demonstrate the effectiveness of the Hawkeye method in proposed IasDig task and dataset, and the practicality for real-world applications.

**Strengths:**

- The overall pipeline of this paper, utilizing large language models, is reasonable and has yielded promising experimental results.
- The visualization of experiments in this paper contributes to a better understanding of the method design and experimental results.

**Limitations:**

- The method proposed in this paper, which includes introducing two feature extraction networks and implementing a two-stage process, potentially suffers from high complexity and low efficiency. Improving final performance might come with significant additional computational costs. I expect the authors to discuss this issue in the rebuttal.
- This paper does not clearly describe the IasDig dataset constructed for studying the video large language model. It lacks detailed introductions of video instances, language information, and comparative analyses with datasets used by other video language models.

**Suitability:**

3

---

### Meta-Review · Area_Chair_9sJ1 · 2024-06-25

**Recommendation:** Accept (Oral)
**Confidence:** 5

**Metareview:**

The reviewers have come up with the following strengths and limitations of the paper

STRENGTH
- Reasonable Pipeline with Promising Results
- Effective Visualization
- Novel Task and Dataset, and Extensive Experiments on Custom-Built Datasets
- Clear and Well-Written
- Sufficient Ablation Experiments
- Innovative Use of Implicit Scene Information
- New Research Direction with IasDig Task
- Integration of Graph-Structured Scene Modeling
- Comprehensive Evaluation

LIMITATIONS
- High Complexity and Low Efficiency
- Lack of Detailed Dataset Description
- Similarity to Existing Tasks (Video Grounding)
- Limited Comparisons, Lack of Comparison with Traditional Methods
- Insufficient Implementation Details
- Generalization and Real-World Validation
- Technical Detail Deficiency

The rebuttal did not change the rating of the reviewers. There is a high consensus among reviewers' rating. The average rating is 4.0.